# Isolation, Identification and Drug Resistance Rates of Bacteria from Pigs in Zhejiang and Surrounding Areas during 2019–2021

**DOI:** 10.3390/vetsci10080502

**Published:** 2023-08-03

**Authors:** Xiangfei Xu, Junxing Li, Pan Huang, Xuemei Cui, Xuefeng Li, Jiaying Sun, Yee Huang, Quanan Ji, Qiang Wei, Guolian Bao, Yan Liu

**Affiliations:** 1Institute of Animal Husbandry and Veterinary Medicine, Zhejiang Academy of Agricultural Sciences, Hangzhou 310021, China; 2020608021037@stu.zafu.edu.cn (X.X.); lijunxing@zaas.ac.cn (J.L.); panhuang@zjnu.edu.cn (P.H.); maycui@zju.edu.cn (X.C.); xuefengli0909@163.com (X.L.); jiquanan@zaas.ac.cn (J.S.); julieyee2017@outlook.com (Y.H.); jiquanan@hotmail.com (Q.J.); weiqiang@zaas.ac.cn (Q.W.); 2College of Veterinary Medicine, Huazhong Agricultural University, Wuhan 430070, China

**Keywords:** bacterial disease, isolation, identification, drug resistance

## Abstract

**Simple Summary:**

This study aimed to determine the prevalence of bacterial diseases in pig farms in various regions of Zhejiang Province and surrounding areas. A total of 526 samples were collected from 85 pig farms in Zhejiang Province and surrounding areas. In this study, samples were analyzed using bacterial isolation and purification, Gram staining, PCR amplification, and antimicrobial susceptibility testing. The isolated bacteria were mainly as follows: *Pasteurella multocida* (*Pm*), *Bordetella bronchiseptica* (*Bb*), *Glasserella parasuis* (*G. parasuis*), *Escherichia coli* (*E. coli*), *Streptococcus suis* (*SS*), and *Actinobacillus pleuropneumoniae* (*APP*). The numbers of bacterial isolates increased during the study period, and mixed infections were observed. Antimicrobial susceptibility testing showed that the drug resistance rates of the various bacteria were high, and the drug resistance spectra were broad. In this study, bacterial epidemiological surveys were conducted on pig farms in various cities in Zhejiang and parts of Anhui, and antimicrobial susceptibility testing was conducted in the isolated strains to provide a scientific basis for the epidemiological and scientific prevention of bacterial diseases, which could be influenced by drug resistance patterns. This study provides scientific guidance for the clinical treatment of bacterial diseases.

**Abstract:**

This study aimed to determine the prevalence of bacterial diseases in pig farms in various regions of Zhejiang Province and surrounding areas. A total of 526 samples were collected from 85 pig farms in Zhejiang Province and surrounding areas. In this study, samples were analyzed using bacterial isolation and purification, Gram staining, PCR amplification, and antimicrobial susceptibility testing. A total of 36 *Pasteurella multocida* (*Pm*) isolates were detected, with an isolation rate of 6.84%; 37 *Bordetella bronchiseptica* (*Bb*) isolates were detected, with an isolation rate of 7.03%; 60 *Glasserella parasuis* (*G. parasuis*) isolates were detected, with an isolation rate of 11.41%; 170 *Escherichia coli* (*E. coli*) isolates were detected, with an isolation rate of 32.32%; 67 *Streptococcus suis* (*SS*) isolates were detected, with an isolation rate of 12.74%; 44 *Actinobacillus pleuropneumoniae* (*APP*) isolates were detected, with an isolation rate of 8.37%; and 7 *Salmonella enteritis* (*SE*) isolates were detected, with an isolation rate of 1.33%. Antimicrobial drug susceptibility testing against 21 types of antibiotics was carried out on the isolated strains, and the results showed that 228 strains had varying degrees of resistance to 21 antibiotics, including *Pm*, *Bb*, *E. coli*, and *APP*, with the highest resistance to lincomycin, at 100%. *Pm* and *APP* were the most sensitive to cephalothin, with resistance rates of 0. In terms of strains, *Pm* had the highest overall sensitivity to 21 antibiotics, and *E. coli* had the highest resistance. In short, bacterial diseases in Zhejiang and the surrounding areas were harmful, and the drug resistance situation was severe. This study provides scientific guidance for the clinical treatment of bacterial diseases.

## 1. Introduction

With the rapid development of large-scale pig farming and the emergence of large-scale circulation of livestock products, the prevalence and threat of bacterial diseases have increased due to the reemergence of old diseases, the emergence of new diseases, the increasing prevalence of mixed infections, increasing drug resistance rates, and gradually expanding drug resistance spectra. The major porcine bacterial diseases can be divided into bacterial infections of the respiratory system and intestinal system. The bacteria that cause respiratory diseases are mainly *Pm*, *Bb* and *APP* [1,2,3,4], and the bacteria that cause intestinal diseases are mainly *E. coli* and *SE* [5,6]. In the past, antibiotics were the main means of treating swine bacterial diseases. In China, the average amount of antibiotics used in veterinary medicine is more than 6000 tons per year [7]. Although antibiotics can prevent bacterial diseases, the abuse of antibiotics also brings a series of problems, such as the increase in drug resistance and environmental pollution [8,9]. According to the literature, previous epidemiological investigations of bacterial diseases in Zhejiang have mainly focused on single-pathogen infections. Previous reports lack a systematic investigation of bacterial diseases. In this study, samples were collected from pig farms in Zhejiang Province and parts of Anhui, with emergency sampling on affected pig farms when necessary, to isolate and identify pathogenic bacteria and clarify the epidemiological characteristics of major bacterial diseases in pigs. Disc diffusion testing of the isolated bacteria for the detection of drug resistance, data to guide bacterial infection treatment is provided.

## 2. Materials and Methods

### 2.1. Main Reagent

From January 2019 to September 2021, the isolation and identification of bacterial pathogens in 526 samples from 85 pig farms submitted from various cities in Zhejiang and parts of Anhui were performed. Antibiotic susceptibility paper sheets were purchased from Changde Bikman Biotechnology Co., Ltd. (Hunan, China) (Table 1).

Green Tap Mix enzyme was purchased from Nanjing Novazan Biotechnology Co., Ltd. (Nanjing, China). DL 2000 DNA Marker was purchased from Takara Biomedical Technology (Beijing, China) Co., Ltd. MacConkey, Tryptic Soy Agar (TSA), and Tryptic Soy Broth (TSB) media were purchased from Thermo Fisher Scientific. Calf serum was purchased from Zhejiang Tianhang Biotechnology Co., Ltd. (Huzhou, China). Nicotinamide adenine dinucleotide (NAD) was purchased from Beijing Soleibo Technology Co., Ltd. (Beijing, China).

### 2.2. Sample Collection

A total of 169 samples were collected or received over a 3-year period, and a total of 526 samples were inspected. The samples were obtained from various organs from dead pig carcasses and effusion, anal swabs, nasal swabs, vaginal swabs, heart blood, fecal samples, environmental samples, etc. Scissors were used to cut the organ surface, and plates were inoculated using an inoculating loop. Effusion, various swab, fecal, environmental, and other samples were placed in a centrifuge tube. PBS was added, the suspension was mixed, and the samples were inoculated onto plate with an inoculation loop and incubated at 37 °C. A single colony was picked from the plate and placed on a new plate, cultured at 37 °C, and purified for 1–2 generations until a single colony grew on the plate. A single colony was picked from the plate and cultured in liquid medium; Gram staining was performed, and the morphological structure of the bacteria was observed under a microscope for preliminary identification and judgment.

### 2.3. PCR Primer

We amplified bacterial-specific genes according to the PCR primers given in the references in Table 2. Primers were synthesized by Suzhou Jinweizhi Biotechnology Co., Ltd. (Suzhou, China).

### 2.4. PCR Template Preparation

A single colony with certain colony characteristics was identified by Gram staining and inoculated into liquid medium. The suspension was placed on a shaker at 37 °C for culture for 8 h. One milliliter of sterilizing liquid, 400 µL of lysis buffer (0.1 mM Tris pH 8.5), and 8 µL of proteinase K were added and mixed well in an EP tube. The sample was incubated at 56 °C for 40 min and then at 100 °C for 10 min. The temperature was decreased to room temperature to preserve the PCR template, and the samples were stored at 4 °C thereafter.

### 2.5. PCR Amplification

The reaction system was as follows: SYBR Green Master Mix 25 µL, template 1 µL, upstream primer 1.5 µL, downstream primer 1.5 µL, added Distilled water up to 50 µL. A total of 30 cycles were run, the annealing temperature was set to 58 °C, and the annealing time was set to 30 s. The PCR products were electrophoresed on a 1.5% agarose gel, and the bands were observed under a UV gel imaging system. Correctly identified strains were cultured in liquid medium to the logarithmic phase. When PCR amplification results were inconsistent with phenotypic results, strains were further detected by 16sRNA sequencing [4,17]. Bacterial liquid and glycerin were added to the EP tube at a ratio of 7 to 3, mixed evenly, and stored at −80 °C.

### 2.6. Antimicrobial Susceptibility Testing

The preserved isolated strains were inoculated on a plate and cultivated overnight at 37 °C. A single colony was picked and inoculated in liquid medium, cultivated to the logarithmic phase at 37 °C and 200 r/min, and subjected to disc diffusion testing. The diameter of the inhibition zone was measured [18].

## 3. Results

### 3.1. Sample Collection

The 169 batches usually included samples from several pigs or samples from multiple parts of a pig, and 526 samples were actually tested. After separation, purification, Gram staining, and PCR amplification, 16 of the tested samples were contaminated and could not be separated and purified, and no bacteria was detected in 89 samples. In the remaining samples, 36 *Pasteurella multocida* (*Pm*) isolates were detected, with an isolation rate of 6.84%; 37 *Bordetella bronchiseptica* (*Bb*) isolates were detected, with an isolation rate of 7.03%; 60 *Glasserella parasuis* (*G. parasuis*) isolates were detected, with an isolation rate of 11.41%; 170 *Escherichia coli* (*E. coli*) isolates were detected, with an isolation rate of 32.32%, *E. coli* isolated in this investigation included ETEC, EPEC, STEC, among which ETEC accounted for the vast majority; 67 *Streptococcus suis* (*SS*) isolates were detected, with an isolation rate of 12.74%; 44 *Actinobacillus pleuropneumoniae* (*APP*) isolates were detected, with an isolation rate of 8.37%; and 7 *Salmonella enteritis* (*SE*) isolates were detected, with an isolation rate of 1.33%(Figure 1A,B). *SE* was not included in the subsequent statistical analysis due to the small number of isolates.

### 3.2. Results of Mixed Infections

Among the various bacteria isolated, there were single infections, double infections, and triple infections. The numbers of single infections, double infections, and triple infections involving *Pm* were 12, 20, and 4, and the proportions of these infections were 33.33%, 55.56%, and 11.11%, respectively. The numbers of single infections, double infections and triple infections involving *Bb* were 18, 15, and 4, and the proportions of these infections were 48.65%, 40.54%, and 10.81%, respectively. The numbers of single infections, double infections and triple infections of *G. parasuis* were 26, 27 and 7, and the proportions of these infections were 43.33%, 46%, 11.67%, respectively. The numbers of single infections, double infections, and triple infections involving *E. coli* were 142, 27, and 1, and the percentages of these infections were 78.89%, 15%, and 0.56%, respectively. The numbers of single infections, double infections and triple infections involving *SS* were 36, 25, and 6, and these infections accounted for 53.73%, 37.31%, and 8.96%, respectively. The numbers of single infections, double infections and triple infections involving *APP* were 38, 5, and 1, and the percentages of these infections were 88.64%, 9.09%, and 2.27%, respectively. *SS* and *G. parasuis* accounted for the largest number of mixed infections, identified in 15 samples. The second largest number of mixed infections (10) involved *Bb* and *G. parasuis* (Table 3 and Table 4, Figure 2).

### 3.3. Results of Disc Diffusion Testing

Disc diffusion testing was performed on the detected strains. The same type of bacteria detected in the same batch in the same field was tested only once, and disc diffusion testing against 21 antibiotics was performed for 228 isolated strains, including 18 strains of *Pm*, 28 strains of *Bb*, 36 strains of *G. parasuis*, 94 strains of *E. coli*, 44 strains of *SS*, and 28 strains of *APP*.

Eighteen isolated *Pm* strains were tested against 21 antibacterial drugs (the numbers of resistant strains, the proportion of drug resistance to total isolates): ampicillin (3 16.67%), amoxicillin (2, 11.11%), penicillin (3, 16.67%), cephalothin (0, 0.00), ceftiofur (1, 5.56%), cefotaxime (0, 0.00), streptomycin (10, 55.56%), gentamicin (9, 50%), amikacin (5, 27.78%), kanamycin (8, 44.44%), spectinomycin (1, 5.56%), apramycin (0, 0.00), erythromycin (1, 5.56%), tilmicosin(14, 77.78%), doxycycline (0, 0.00), tetracycline (5, 27.78%), florfenicol (2, 11.11%), lincomycin (18, 100%), compound trimethoprim (11, 61.11%), enrofloxacin (2, 11.11%), and ciprofloxacin (2, 11.11%). The results reveal that *Pm* was most resistant to lincomycin, with a resistance rate of 100%, and it was sensitive to cephalothin, cefotaxime, apramycin, and doxycycline, with sensitivity rates of 100%. There were five antibiotics associated with a resistance rate of 50% or above: streptomycin (55.56%), gentamicin (50%), tilmicosin (77.78%), lincomycin (100%), and compound trimethoprim (61.11%). In terms of drug types, the highest sensitivity was observed for cephalosporins, the highest resistance rate was observed for ceftiofur, at only 5.56%, and the cephalothin and cefotaxime resistance rates were 0.00%. Among the aminoglycosides, the streptomycin and gentamicin resistance rates were 55.56% and 50%, respectively, and the apramycin resistance rate was 0%. Among the tetracyclines, the tilmicosin resistance rate was 77.78%, and the doxycycline sensitivity rate was 100%. (Table 5).

A total of 28 isolated *Bb* strains were tested against 21 antibacterial drugs (the numbers of resistant strains, the proportion of drug resistance to total isolates): ampicillin (27, 96.42%), amoxicillin (13, 46.43%), penicillin (27, 96.42%), cephalothin (13, 46.43%), ceftiofur (20, 71.42%), cefotaxime (17, 60.71%), streptomycin (26, 92.86%), gentamicin (9, 32.14%), amikacin (9, 32.14%), kanamycin (9, 32.14%), spectinomycin (21, 75%), apramycin (27, 96.42%), erythromycin (13, 21.43%), tilmicosin (26, 92.86%), doxycycline (6, 28.57%), tetracycline (8, 46.43%), florfenicol (13, 42.86%), lincomycin (28, 100%), compound trimethoprim (27, 96.42%), enrofloxacin (12, 46.43%), and ciprofloxacin (5, 17.86%). The results show that *Bb* was the most resistant to lincomycin, with a resistance rate of 100%. Ten antibiotics had associated resistance rates of 50% or higher: ampicillin (96.42%), penicillin (96.42%), ceftiofur (71.42%), cefotaxime (60.71%), streptomycin (92.86%), apramycin (96.42%), tilmicosin (92.86%), lincomycin (100%), and compound trimethoprim (96.42%). Among the quinolones, ciprofloxacin was associated with the highest sensitivity, with an associated resistance rate of 17.86% (Table 6).

A total of 36 isolated *HPS* strains were tested against 21 antibacterial drugs (the numbers of resistant strains, the proportion of drug resistance to total isolates): ampicillin (20, 55.56%), amoxicillin (14, 38.89%), penicillin (17, 47.22%), cephalothin (7, 19.44%), ceftiofur (8, 22.22%), cefotaxime (3, 8.33%), streptomycin (24, 66.67%), gentamicin (20, 55.56%), amikacin (22, 61.11%), kanamycin (13, 36.11%), spectinomycin (5, 13.89%), apramycin (33, 91.67%), erythromycin (16, 44.44%), tilmicosin (25, 69.44%), doxycycline (2, 5.56%), tetracycline (9, 25%), florfenicol (7, 19.44%), lincomycin (30, 83.33%), compound trimethoprim (34, 94.44%), enrofloxacin (18, 50%), and ciprofloxacin (11, 30.56%). The results reveal that *HPS* was the most resistant to compound trimethoprim, with a resistance rate of 94.44%, and most sensitive to doxycycline, with a resistance rate of 5.56%. Seven antibiotics were associated with resistance rates of 50% or higher: ampicillin (55.56%), penicillin (60.71%), streptomycin (66.67%), gentamicin (55.56%), amikacin (61.11%), apramycin (91.67%), and tilmicosin (69.44%). The following cephalosporin antibiotics were associated with the highest sensitivity, with drug resistance rates of less than 25%: cephalothin (19.44%), ceftiofur (22.22%), and cefotaxime (8.33%) (Table 7).

Ninety-four isolated *E. coli* strains were tested against 21 antibacterial drugs (the numbers of resistant strains, the proportion of drug resistance to total isolates): ampicillin (87, 92.55%), amoxicillin (84, 89.36%), penicillin (92, 97.87%), cephalothin (55, 58.51%), ceftiofur (30, 31.91%), cefotaxime (15, 15.96%), streptomycin (58, 61.70%), gentamicin (62, 65.96%), amikacin (25, 26.60%), kanamycin (52, 55.32%), spectinomycin (31, 32.98%), apramycin (53, 56.38%), erythromycin (79, 84.04%), tilmicosin (94, 100%), doxycycline (86, 91.49%), tetracycline (88, 93.62%), florfenicol (69, 73.40%), lincomycin (94, 100%), compound trimethoprim (86, 91.49%), enrofloxacin (71, 75.53%), and ciprofloxacin (44, 46.80%). The results reveal that *E. coli* was most resistant to tilmicosin and lincomycin, with resistance rates reaching 100%. Sensitivity to cefotaxime was the highest, and the resistance rate was 15.96%. There were 15 antibiotics associated with resistance rates of 50% or higher: ampicillin (92.55%), amoxicillin (89.36%), penicillin (97.87%), cephalothin (58.51%), streptomycin (61.70%), gentamicin (65.96%), kanamycin (55.32%), apramycin (56.38%), erythromycin (84.04%), tilmicosin (100%), doxycycline (91.49%), tetracycline (93.62%), florfenicol (73.40%), lincomycin (100%), and enrofloxacin (75.53%). Cephalosporins were associated with the highest sensitivity, with associated resistance rates of 58.51% against cephalothin, 31.91% against ceftiofur, and 15.96% against cefotaxime. In general, *E. coli* exhibited the highest level of antibiotic resistance (Table 8).

Forty-four isolated *SS* strains were tested against 21 antibacterial drugs (the numbers of resistant strains, the proportion of drug resistance to total isolates): ampicillin (10, 22.72%), amoxicillin (3, 6.82%), penicillin (6, 13.64%), cephalothin (1, 2.27%), ceftiofur (3, 6.82%), cefotaxime (4, 9.09%), streptomycin (26, 59.09%), gentamicin (40, 90.91%), amikacin (43, 97.73%), kanamycin (41, 93.18%), spectinomycin (16, 36.36%), apramycin (44, 100%), erythromycin (30, 68.18%), tilmicosin (43, 97.73%), doxycycline (14,31.82%), tetracycline (37, 84.09%), florfenicol (8, 18.08%), lincomycin (43, 97.73%), compound trimethoprim (33, 75%), enrofloxacin (19, 43.18%), and ciprofloxacin (14, 31.82%). The results show that *SS* was most resistant to apramycin, with a resistance rate of 100%, and had the highest sensitivity to cephalothin, with a resistance rate of 2.27%. There were nine antibiotics associated with resistance rates of 50% or higher: streptomycin (59.09%), gentamicin (90.91%), amikacin (97.73%), kanamycin (93.18%), apramycin (100%), erythromycin (68.18%), tilmicosin (97.73%), tetracycline (84.09%), and lincomycin (97.73%). SS had the highest sensitivity to cephalosporin antibiotics. The cephalothin resistance rate was 2.27%, the ceftiofur resistance rate was 6.82%, and the cefotaxime resistance rate was 9.09% (Table 9).

Twenty-eight isolated *APP* strains were tested against 21 antibacterial drugs (the numbers of resistant strains, the proportion of drug resistance to total isolates): ampicillin (11, 39.29%), amoxicillin (6, 21.42%), penicillin (13, 46.42%), cephalothin (1, 3.57%), ceftiofur (2, 7.14%), cefotaxime (0, 0.00), streptomycin (15, 53.57%), gentamicin (8, 28.57%), amikacin (15, 53.57%), kanamycin (15, 53.57%), spectinomycin (6, 21.42%), apramycin (24, 85.71%), erythromycin (3, 10.71%), tilmicosin(24, 85.71%), doxycycline (4, 14.28%), tetracycline (13, 46.42%), florfenicol (7, 25.00%), lincomycin (27, 96.43%), compound trimethoprim (16, 57.14%), enrofloxacin (11, 39.29%), and ciprofloxacin (0, 0). The results reveal that *APP* was most resistant to lincomycin, with a resistance rate of 96.43%, and most sensitive to cephalothin and ciprofloxacin, with resistance rates of 0. There were six antibiotics with associated resistance rates of 50% or higher: streptomycin (53.57%), amikacin (53.57%), kanamycin (53.57%), tilmicosin (85.71%), lincomycin (96.43%), and compound trimoxazole (57.14%). Cephalosporin antibiotics were associated with the highest sensitivity, with associated resistance rates of 0% for cephalothin, 7.14% for ceftiofur, and 3.57% for cefotaxime. Two values in Table 10 were ignored because they were zero (Table 10).

### 3.4. Multidrug Resistance of Isolated Strains

Among the 18 strains of *Pm* that were subjected to disc diffusion testing, 6 strains were resistant to 0–5 drugs, accounting for 33.33%; 11 strains were resistant to 6–10 drugs, accounting for 61.11%; 1 strain was resistant to 12 drugs, accounting for 5.56%; and no strains resistant to 16–20 or 21 drugs. Among the 28 *Bb* strains tested, 1 strain was resistant to 0–5 drugs, accounting for 3.57%; 7 strains were resistant to 6–10 drugs, accounting for 25%; 15 strains were resistant to 15 drugs, accounting for 53.57%; 5 strains resistant to 16–20 drugs, accounting for 17.86%; and no strains were resistant to 21 drugs. Among the 36 strains of *G. parasuis* tested, 3 strains were resistant to 0–5 drugs, accounting for 8.33%; 20 strains were resistant to 6–10 drugs, accounting for 55.56%; 12 strains were resistant to 11–15 drugs, accounting for 33.33%; 2 strains were resistant to 16–20 drugs, accounting for 5.56%; and no strains resistant to 21 drugs. Among the 94 *E. coli* strains tested, 1 strain was resistant to 0–5 drugs, accounting for 1.06%; 14 strains were resistant to 6–10 drugs, accounting for 38.89%; 11–15 strains were resistant to 42 drugs, accounting for 44.68%; 31 strains were resistant to 16–20 drugs, accounting for 32.98%; and 6 strains were resistant to 21 drugs, accounting for 6.38%. Among the 44 strains of SS tested, 2 strains were resistant to 0–5 drugs, accounting for 4.55%; 16 strains were resistant to 6–10 drugs, accounting for 36.36%; 18 strains were resistant to 11–15 drugs, accounting for 40.91%; 8 strains were resistant to 16–20 drugs, accounting for 18.18%; there were no strains resistant to 21 drugs. Among the 28 *APP* strains tested, 8 strains were resistant to 0–5 drugs, accounting for 28.57%; 13 strains were resistant to 6–10 drugs, accounting for 29.55%; 7 strains were resistant to 11–15 drugs, accounting for 15.91%; and there were no strains resistant to 16–20 or 21 drugs. The experimental results revealed that none of the isolated strains were resistant to 0 or 1 antibiotic. Pm had the best overall sensitivity, and E. coli had the highest rate of drug resistance. All the bacteria except for *E. coli* were resistant to 19 or more drugs; *E. coli* had resistance to 19 or more drugs, and 6 strains were resistant to all 21 drugs (Table 11, Figure 3).

### 3.5. Heatmap of Drug Resistance

Among the *Pm*, *Bb*, *G. parasuis*, *E. coli*, *SS*, and *APP* isolates that were tested with the disc diffusion method, 18 were randomly selected for construction of a heatmap of resistance to 21 antibiotics (Figure 4). The results showed that *E. coli* was most resistant to several antibiotics. *Pm* and *SS* were similar in their sensitivity rates, and both were sensitive to antibiotics A, B, C, D, E, F, G, H, and I. Four antibiotics, trimethoprim, tilmicosin, apramycin, and lincomycin, had the highest associated resistance rates, which may be related to the high rates of use of these antibiotics on pig farms. *Pm*, *G. parasuis*, *SS*, and *APP* had similar drug resistance patterns. Most of these bacterial specimens were collected from the lungs, and the resistance patterns may be related to the sample collection site.

### 3.6. Three-Year Trend and Seasonal Trend of Bacteria

The year was divided into four seasons: spring, summer, autumn and winter. Spring was defined as February–April, summer was defined as May–July, autumn was defined as August–October, and winter was defined as November–January, and a line chart of bacterial prevalence rates in the four seasons was constructed (Figure 5A). According to the results, the numbers of isolates of the six bacteria were the lowest in autumn and highest in spring, with a decreasing trend from summer to autumn and an increasing trend in winter. Since this experiment was carried out only through September 2021, the three-year bacterial prevalence trend map was created using the isolates collected in the first 9 months of each year during 2019–2021 (Figure 5B). It can be seen in the figure that the number of *E. coli* isolates increased during the three-year period, while the numbers of the other five bacteria all decreased from 2019 to 2020, with upward trends in 2021.

## 4. Discussion

The experimental results revealed that *Pm*, *Bb*, *G. parasuis*, *E coli*, *SS*, and *APP* were widespread in Zhejiang, similar to previous reports [19] in China, though the bacterial isolation rates were different. For example, a previous report [20,21] showed that the isolation rate of *SS* in Thailand was very high, but the isolation rate of *SS* in this investigation was very low. This result may be caused by comprehensive factors, such as different regional characteristics, sample sources, and climates. The bacteria in this study can be divided into two categories, intestinal bacteria and respiratory bacteria, and infections caused by these bacteria can seriously affect the Chinese pig industry. Therefore, research on the isolation rates, drug resistance patterns, and seasonality of these bacteria is of great significance for the prevention and control of local bacterial diseases.

In this experiment, antimicrobial susceptibility testing against 21 antibiotics was carried out in the isolated strains. The results showed that in *Pm*, the resistance rate for lincomycin was the highest, at 100%. Sensitivity to cephalothin, cefotaxime, apramycin, and doxycycline was high, with sensitivity rates of 100%. A previous report [22] showed that *Pm* had the highest resistance to amoxicillin, with a resistance rate of 75.9%, in Vietnam. In this experiment, *Bb* had the highest resistance to lincomycin, with a resistance rate of 100%, and was relatively sensitive to ciprofloxacin, with a resistance rate of 17.86%. Another report [23] showed that *Bb* had the highest resistance to ampicillin, with a resistance rate of 83.98%, and there were 6 antibiotics with associated resistance rates of 0% in China. *E. coli* was had the highest resistance to tilmicosin and lincomycin, with resistance rates of 100%, and the highest sensitivity to cefotaxime, with a resistance rate of 15.96%. A previous report [24] showed that *E. coli* had highest resistance to trimethoprim, with a resistance rate of 100% in Poland. *SS* had the highest resistance to apramycin, with a resistance rate of 100%, and the highest sensitivity to cephalothin, with a resistance rate of 2.27%. Another previous report [25] showed that the highest rate of drug resistance in *Streptococcus* was associated with tetracycline, with a drug resistance rate of 84.2% in Ontario. There was a large gap in the resistance rates of different strains to different drug categories and even different antibiotics in similar drug categories. This is related to different treatment regimens of various countries and regions. This result demonstrates that for the treatment of bacterial diseases, more targeted medications based on drug resistance testing results are needed. In this study, the most common coinfection involved *SS* and *G. parasuis*. Coinfection is often more threatening to pigs than monoinfection [26,27]. None of the isolates of the six bacteria that were subjected to antimicrobial susceptibility testing were resistant to 0 or only 1 antibiotic. Six strains of *E. coli* showed multiresistance to 21 drugs simultaneously. On the basis of these investigation results, the drug resistance situation is severe. Additionally, the resistance patterns of each bacterial species showed certain differences according to antibiotic properties. Seasonal distributions of these bacteria were clearly observed in this study. These bacteria were generally prevalent in spring and began to decline in summer until autumn. In winter, the bacterial prevalence showed an increasing trend. This phenomenon may be related to the fact that hot temperatures in summer are not conducive to the growth of bacteria, while milder temperatures in spring are suitable for bacterial growth. Season affects prevalence of bacterial pathogenic genes [28,29]. Therefore, we cannot ignore the effects of temperature when implementing prevention and control measures for bacteria. The number of *E. coli* isolated during the three-year study period consistently increased, while the numbers of the other five bacteria declined from 2019 to 2020 and then increased in 2021. The monitoring of bacterial diseases in Zhejiang will be indispensable in the next few years. Future research will require long-term monitoring and bacterial typing to effectively guide prevention, control and treatment measures targeting bacteria.

## 5. Conclusions

This study focused on the isolation and identification of bacterial pathogens in 516 samples from 85 pig farms submitted for inspection from various cities in Zhejiang and parts of Anhui. The isolated bacteria were mainly as follows: *Pm*, *Bb*, *G. parasuis*, *E. coli*, *SS*, and *APP*. The numbers of bacterial isolates increased during the study period, and mixed infections were observed. Antimicrobial susceptibility testing showed that the drug resistance rates of the various bacteria were high, and the drug resistance spectra were broad. In this study, bacterial epidemiological surveys were conducted on pig farms in various cities in Zhejiang and parts of Anhui, and antimicrobial susceptibility testing was conducted in the isolated strains to provide a scientific basis for the epidemiological and scientific prevention of bacterial diseases, which could be influenced by drug resistance patterns. This study provides scientific guidance for the clinical treatment of bacterial diseases.

## Figures and Tables

**Figure 1 vetsci-10-00502-f001:**
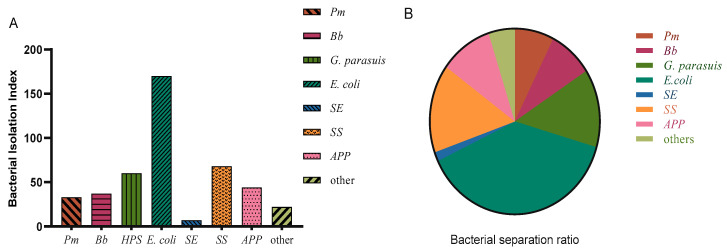
(**A**) Bacterial isolation results and (**B**) isolation rates.

**Figure 2 vetsci-10-00502-f002:**
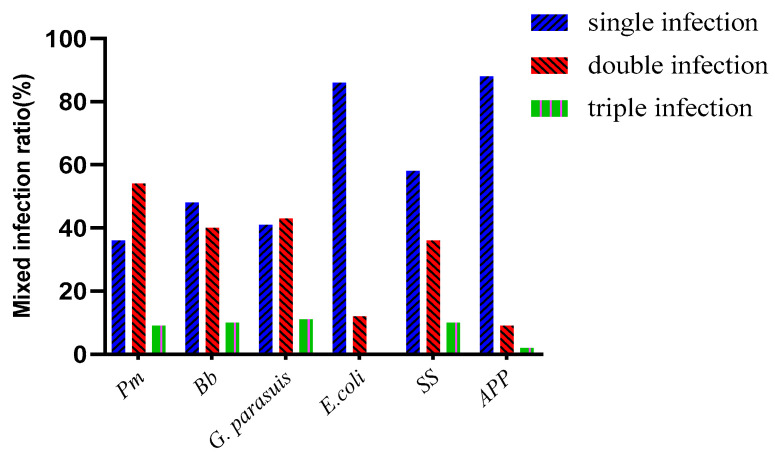
Mixed infection results (%).

**Figure 3 vetsci-10-00502-f003:**
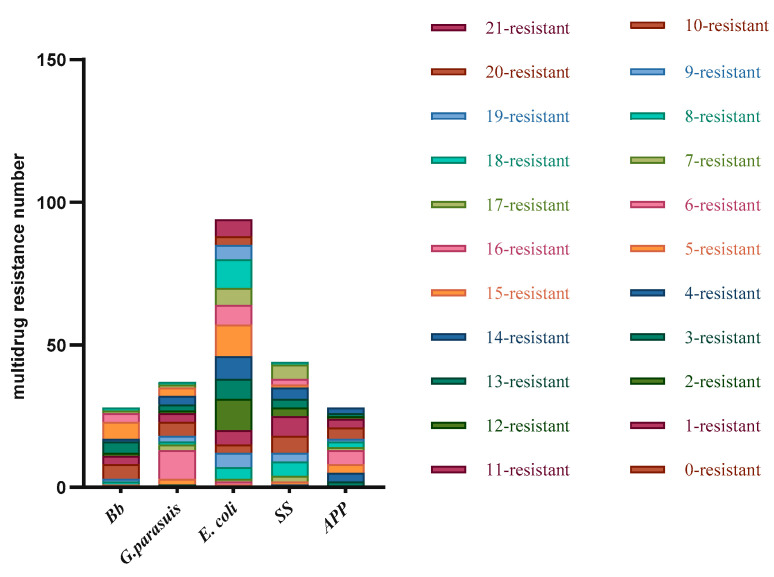
Multidrug resistance results of the six bacteria.

**Figure 4 vetsci-10-00502-f004:**
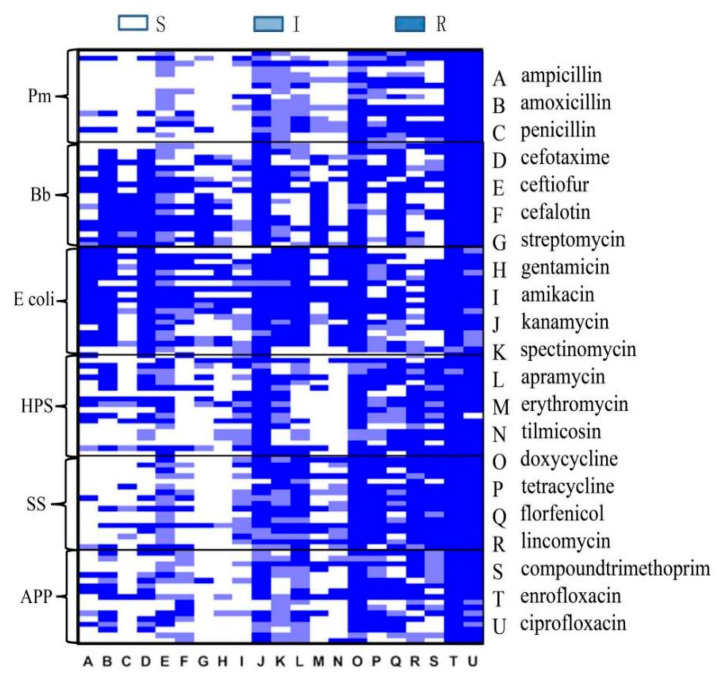
Heatmap of resistance.

**Figure 5 vetsci-10-00502-f005:**
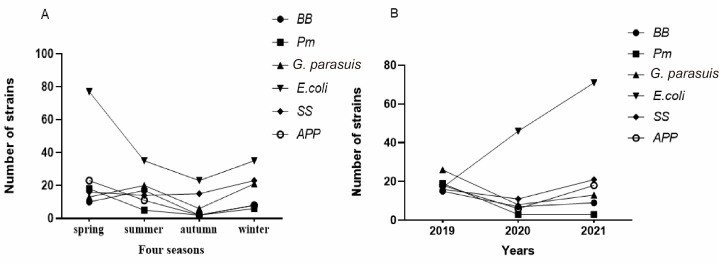
(**A**) Seasonal trends of bacteria and (**B**) three-year trend of bacterial prevalence.

**Table 1 vetsci-10-00502-t001:** Types of antibiotic susceptibility discs.

Antibiotic Category	Drug Name
penicillins	ampicillin, amoxicillin, penicillin
cephalosporins	cefotaxime, ceftiofur, cephalothin
aminoglycosides	streptomycin, gentamicin, amikacin, apramycin, kanamycin, spectinomycin
macrolides	tilmicosin, erythromycin
tetracyclines	doxycycline, tetracycline
chloramphenicol	florfenicol
lincomycins	lincomycin
sulfonamides	compound trimethoprim
quinolones	enrofloxacin, ciprofloxacin

**Table 2 vetsci-10-00502-t002:** Primers used in study.

Strains	Gene	Name	Sequence(5′→3′)	Amplicons Size	References
*Pasteurella multocida*	*ToxA*	ToxA–1	CTTAGATGAGCGACAAGG	846 bp	[10]
		ToxA–2	GAATGCCACACCTCTATAG		
*Bordetella bronchiseptica*	Fla	Fla4	TGGCGCCTGCCCTATC	237 bp	[11]
		Fla2	AGGCTCCCAAGAGAGAAAGGCTT		
*Glasserella parasuis*	16SrRNA	HPS–1	GGCTTCGTCACCCTCTGT	822 bp	[12]
		HPS–2	GTGATGAGGAAGGGTGGTGT		
*Escherichia coli*	uidA	Ec–1	AAAACGGCAAGAAAAAGCAG	147 bp	[13]
		Ec–2	GCGTGGTTACAGTCTTGCG		
*Streptococcus suis*	*gdh*	JP4	GCA GCGTATTCTGTCAAACG	689 bp	[14]
		JP5	CCATGGACAGATAAA GATGG		
*Actinobacillus pleuropneumoniae*	apxIVA	APXIVA–1	TGGCACTGACGGTGATGA	442 bp	[15]
		APXIVA–2	GGCCATCGACTCAACCAT		
*Salmonella enteritis*	*invA*	139	GTGAAATTATCGCCACGTTCGGGCAA	284 bp	[16]
		141	TCATCGCACCGTCAAAGGAACC		

**Table 3 vetsci-10-00502-t003:** Mixed infection results.

Strain Name	Single Infection	Double Infection	Triple Infection
*Pm*	12	20	4
*Bb*	18	15	4
*G. parasuis*	26	27	7
*E. coli*	142	27	1
*SS*	36	25	6
*APP*	39	4	1

**Table 4 vetsci-10-00502-t004:** Number of mixed infections.

	Double Infection		Triple Infection	
	Types	Number	Types	Number
*Pm*	*Pm* + *Bb*	1	*Pm* + *APP* + others	1
	*Pm* + *G. parasuis*	5	*Pm* + *G. parasuis* + *SS*	3
	*Pm* + *E. coli*	1		
	*Pm* + *SS*	3		
	*Pm* + *APP*	2		
	*Pm* + *others*	3		
*Bb*	*Bb* + *Pm*	1	*Bb* + *G. parasuis* + *E.coli*	1
	*Bb* + *G. parasuis*	6	*Bb* + *G. parasuis* + *SS*	3
	*Bb* + *SS*	4		
	*Bb* + *E.coli*	3		
	*Bb + others*	3		
*G. parasuis*	*G. parasuis* + *Pm*	5	*G. parasuis* + *SS* + *Bb*	3
	*G. parasuis* + *Bb*	6	*G. parasuis* + *SS* + *Pm*	3
	*G. parasuis* + *SS*	9	*G. parasuis* + *E.coli* + *Bb*	1
	*G. parasuis* + *E.coli*	3	*G. parasuis* + *Pm* + *others*	1
	*G. parasuis* + *APP*	1		
	*G. parasuis* + *others*	4		
*E.coli*	*E.coli* + *Pm*	1	*E.coli* + *G. parasuis* + *Bb*	1
	*E.coli* + *Bb*	3		
	*E.coli* + *G. parasuis*	3		
	*E.coli* + *SS*	8		
	*E.coli* + *others*	1		
*SS*	*SS* + *Bb*	4	*SS* + *Pm* + *G. parasuis*	3
	*SS* + *Pm*	3	*SS* + *Bb* + *G. parasuis*	3
	*SS* + *G. parasuis*	9		
	*SS* + *E.coli*	8		
	*SS* + *APP*	1		
*APP*	*APP* + *Pm*	2	*APP* + *Pm* + *others*	1
	*APP* + *G. parasuis*	1		
	*APP* + *SS*	1		

**Table 5 vetsci-10-00502-t005:** *Pm* isolate disc diffusion testing results.

Types of Antibiotics	Antibiotic Name	Number of Strains			Resistance Rate (%)
		S	I	R	
penicillins	ampicillin	15	0	3	16.67
	amoxicillin	15	1	2	11.11
	penicillin	15	0	3	16.67
cephalosporins	cefotaxime	18	0	0	0.00
	ceftiofur	16	1	1	5.56
	cephalothin	18	0	0	0.00
aminoglycosides	streptomycin	3	5	10	55.56
	gentamicin	1	8	9	50
	amikacin	10	3	5	27.78
	kanamycin	4	6	8	44.44
	spectinomycin	12	5	1	5.56
	apramycin	18	0	0	0.00
macrolides	tilmicosin	0	4	14	77.78
	erythromycin	2	15	1	5.56
tetracyclines	doxycycline	13	5	0	0.00
	tetracycline	5	8	5	27.78
chloramphenicol	florfenicol	12	4	2	11.11
lincomycins	lincomycin	0	0	18	100
sulfonamides	compound trimethoprim	1	6	11	61.11
quinolones	enrofloxacin	5	11	2	11.11
	ciprofloxacin	14	2	2	11.11

S: susceptible, I: intermediate, and R: resistance.

**Table 6 vetsci-10-00502-t006:** *Bb* isolate disc diffusion test results.

Types of Antibiotics	Antibiotic Name	Number of Strains			Resistance Rate (%)
		S	I	R	
penicillins	ampicillin	0	1	27	96.42
	amoxicillin	11	4	13	46.42
	penicillin	1	0	27	96.42
cephalosporins	cefotaxime	11	4	13	46.42
	ceftiofur	6	2	20	71.42
	cephalothin	8	3	17	60.71
aminoglycosides	streptomycin	2	0	26	92.86
	gentamicin	17	2	9	32.14
	amikacin	15	4	9	32.14
	kanamycin	15	4	9	32.14
	spectinomycin	7	0	21	75
	apramycin	1	0	27	96.42
macrolides	erythromycin	10	5	13	46.42
	tilmicosin	0	2	26	92.86
tetracyclines	doxycycline	17	5	6	21.43
	tetracycline	14	6	8	28.57
chloramphenicol	florfenicol	13	2	13	46.43
lincomycins	lincomycin	0	0	28	100
sulfonamides	compound trimethoprim	0	1	27	96.42
quinolones	enrofloxacin	4	12	12	42.86
	ciprofloxacin	15	8	5	17.86

S: susceptible, I: intermediate, and R: resistance.

**Table 7 vetsci-10-00502-t007:** *G. parasuis* isolate disc diffusion test results.

Types of Antibiotics	Antibiotic Name	Number of Strains			Resistance Rate (%)
		S	I	R	
penicillins	ampicillin	11	5	20	55.56
	amoxicillin	22	0	14	38.89
	penicillin	17	2	17	47.22
cephalosporins	cefotaxime	27	2	7	19.44
	ceftiofur	25	3	8	22.22
	cephalothin	26	7	3	8.33
aminoglycosides	streptomycin	7	5	24	66.67
	gentamicin	11	5	20	55.56
	amikacin	7	7	22	61.11
	kanamycin	7	16	13	36.11
	spectinomycin	28	3	5	13.89
	apramycin	1	2	33	91.67
macrolides	erythromycin	12	8	16	44.44
	tilmicosin	3	10	25	69.44
tetracyclines	doxycycline	34	0	2	5.56
	tetracycline	19	8	9	25
chloramphenicol	florfenicol	28	1	7	19.44
lincomycins	lincomycin	1	5	30	83.33
sulfonamides	compound trimethoprim	0	2	34	94.44
quinolones	enrofloxacin	10	8	18	50
	ciprofloxacin	9	16	11	30.56

S: susceptible, I: intermediate, and R: resistance.

**Table 8 vetsci-10-00502-t008:** Disc diffusion test results for *E. coli* isolates.

Types of Antibiotics	Antibiotic Name	Number of Strains			Resistance Rate (%)
		S	I	R	
penicillins	ampicillin	2	5	87	92.55
	amoxicillin	9	1	84	89.36
	penicillin	2	0	92	97.87
cephalosporins	cefotaxime	33	6	55	58.51
	ceftiofur	53	11	30	31.91
	cephalothin	52	27	15	15.96
aminoglycosides	streptomycin	27	9	58	61.70
	gentamicin	26	6	62	65.96
	amikacin	53	16	25	26.60
	kanamycin	23	19	52	55.32
	spectinomycin	55	8	31	32.98
	apramycin	6	35	53	56.38
macrolides	erythromycin	0	15	79	84.04
	tilmicosin	0	0	94	100
tetracyclines	doxycycline	7	1	86	91.49
	tetracycline	3	3	88	93.62
chloramphenicol	florfenicol	23	2	69	73.40
lincomycins	lincomycin	0	0	94	100
sulfonamides	compound trimethoprim	1	7	86	91.49
quinolones	enrofloxacin	6	17	71	75.53
	ciprofloxacin	31	15	44	46.80

S: susceptible, I: intermediate, and R: resistance.

**Table 9 vetsci-10-00502-t009:** *SS* isolate disc diffusion test results.

Types of Antibiotics	Antibiotic Name	Number of Strains			Resistance Rate (%)
		S	I	R	
penicillins	ampicillin	30	4	10	22.72
	amoxicillin	40	1	3	6.82
	penicillin	35	3	6	13.64
cephalosporins	cefotaxime	41	2	1	2.27
	ceftiofur	39	2	3	6.82
	cephalothin	34	6	4	9.09
aminoglycosides	streptomycin	14	4	26	59.09
	gentamicin	3	1	40	90.91
	amikacin	0	1	43	97.73
	kanamycin	1	2	41	93.18
	spectinomycin	23	5	16	36.36
	apramycin	0	0	44	100
macrolides	erythromycin	1	13	30	68.18
	tilmicosin	1	0	43	97.73
tetracyclines	doxycycline	14	16	14	31.82
	tetracycline	3	4	37	84.09
chloramphenicol	florfenicol	27	9	8	18.18
lincomycins	lincomycin	1	0	43	97.73
sulfonamides	compound trimethoprim	1	10	33	75
quinolones	enrofloxacin	8	17	19	43.18
	ciprofloxacin	13	17	14	31.82

S: susceptible, I: intermediate, and R: resistance.

**Table 10 vetsci-10-00502-t010:** *APP* isolate disc diffusion test results.

Types of Antibiotics	Antibiotic Name	Number of Strains			Resistance Rate (%)
		S	I	R	
penicillins	ampicillin	13	4	11	39.29
	amoxicillin	19	3	6	21.42
	penicillin	10	5	13	46.42
cephalosporins	cefotaxime	26	2	0	0.00
	ceftiofur	26	0	2	7.14
	cephalothin	26	1	1	3.57
aminoglycosides	streptomycin	9	4	15	53.57
	gentamicin	10	10	8	28.57
	amikacin	7	6	15	53.57
	kanamycin	6	7	15	53.57
	spectinomycin	22	0	6	21.42
	apramycin	0	2	24	85.71
macrolides	erythromycin	8	17	3	10.71
	tilmicosin	1	3	24	85.71
tetracyclines	doxycycline	21	1	4	14.28
	tetracycline	3	12	13	46.42
chloramphenicol	florfenicol	10	11	7	25.00
lincomycins	lincomycin	1	0	27	96.43
sulfonamides	compound trimethoprim	4	8	16	57.14
quinolones	enrofloxacin	11	6	11	39.29
	ciprofloxacin	21	7	0	0.00

S: susceptible, I: intermediate, and R: resistance.

**Table 11 vetsci-10-00502-t011:** Multidrug resistance ratios of the six bacteria.

Number of Antibiotics with Associated Resistance	*Pm*	*Bb*	*G. parasuis*	*E. coli*	*SS*	*APP*
0	0	0	0	0	0	0
1	0	0	0	0	0	0
2	5.88%	0	0	0	0	0
3	11.76%	0	2.70%	0	0	7.14%
4	11.76%	0	0	0	2.27%	10.71%
5	5.88%	3.57%	5.40%	1.06%	2.27%	10.71%
6	17.64%	0	27.03%	1.06%	0	17.82%
7	11.76%	0	5.40%	1.06%	4.54%	3.57%
8	11.76%	3.57%	2.70%	4.25%	11.36%	7.14%
9	5.88%	3.57%	5.40%	5.31%	6.81%	3.57%
10	17.64%	17.86%	13.51%	3.19%	13.63%	14.29%
11	0	10.71%	8.11%	5.31%	15.90%	10.71%
12	5.88%	3.57%	2.70%	11.70%	6.81%	3.57%
13	0	14.29%	5.40%	7.45%	6.81%	3.57%
14	0	3.57%	8.11%	8.51%	9.09%	7.14%
15	0	21.43%	8.11%	11.70%	2.27%	0
16	0	10.71%	0	7.45%	4.54%	0
17	0	3.57%	2.70%	6.38%	11.36%	0
18	0	3.57%	2.70%	10.64%	2.27%	0
19	0	0	0	5.31%	0	0
20	0	0	0	3.19%	0	0
21	0	0	0	6.38%	0	0

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
