# Peer review of "Isolation, Identification and Drug Resistance Rates of Bacteria from Pigs in Zhejiang and Surrounding Areas during 2019–2021"

_vetsci, 2023, doi:10.3390/vetsci10080502_

Round 1

Reviewer 1 Report

This manuscript provides important data regarding the epidemiology and incidence of the major bacterial disease in pigs and their antibiotic resistance. I think that after a major revision, this data should be published.

I come with some suggestions:

-          Line 32: Haemophilus parasuis is not named like this any more it is Glasserella parasuis

-          Line 33: E. coli is to general to say, you need to be more specific

-          Lines 80-81: A total of 169 were collected, but 516 were inspected… were do the other 347 come from?

-          Line 83: were the scissors decontaminated after each use?

-          PCR Protocol: did you use your own method, protocol? Or is it inspired by another paper? Please mention

-          Lines 123-130: it is very hard to follow, try to use several sentences…

-          Fig 1: make them bigger

-          Lines 134-151: make a diagram (Venn Diagram) to point out the different coinfections between bacteria with percentage

-          Lines 165-168, it looks like random numbers with percentages, please use sentences to explain ( same for line 186-189 and so on…)

-          Lines 156-276 (including tables 4-9): The information is just mentioned and enumerated… I think you can join all the important information from each table, and create a new one. Also, write 3-4 paragraphs to explain the resistance. How you did it is too hard to follow

The quality of English is over the average. All the information is written well in common sense, minor changes should apply.

Author Response

Dear Reviewer,

Thank you very much for reviewing our manuscript (vetsci-2495641) and for your valuable comments and suggestions. They are very helpful for improving our manuscript. According to your comments and suggestions, we thoroughly revised our manuscript and submitted it to Antibiotics for further review. The changes based on your suggestions were highlighted in red colour in the new version. Point by point corrections were made as follows:

Line 32: Haemophilus parasuis is not named like this any more it is Glasserella parasuis

Response 1: Thank you very much for your comments. I have changed all Haemophilus parasuis in the manuscript to Glasserella parasuis(G. parasuis). Please see lines 18, 34, 129, 147, 154, 156, 171, 317,345 and 399.

Line 33: E. coli is to general to say, you need to be more specific

Response 2: Thank you. The E. coli isolated by us included ETEC, EPEC, STEC, and most of them were ETEC, ETEC is divided into different adhesins and toxins. We organized it into another paper. I have also added instructions to the manuscript: “E. coli isolated in this investigation included ETEC, EPEC, STEC, among which ECET accounted for the vast majority”. Please see lines 131,132.

Lines 80-81: A total of 169 were collected, but 526 were inspected… were do the other 347 come from?

Response 3: Thank you. 169 refers to 169 batches of samples submitted for examination or epidemiological investigation. For example, a sample sent for inspection may include samples from several pigs or samples from different parts of a pig. So the actual sample size we tested was 526. I have also added in the manuscript. Please see line 124-125.

Line 83: were the scissors decontaminated after each use?

Response 4: Thank you. We will certainly do this, when the sample collection is related to the utensils will be alcohol disinfection or alcohol lamp combustion disinfection.

PCR Protocol: did you use your own method, protocol? Or is it inspired by another paper? Please mention

Response 5: Thank you. The PCR procedure is based on the procedure in the reference where primers are given. At the same time, I also inserted 2 references in the article. Please see lines 108-115.

Lines 123-130: it is very hard to follow, try to use several sentences…

Response 6: Thank you. I have changed this passage in the manuscript. Please see lines 124-137.

Fig 1: make them bigger

Response 7: Thank you. I have already made the adjustment. Please see Figure.1.

Lines 134-151: make a diagram (Venn Diagram) to point out the different coinfections between bacteria with percentage

Response 8: Thank you. We thought that Venn diagram could not represent the data well, so I presented the data in table. Please see table 4.

Lines 165-168, it looks like random numbers with percentages, please use sentences to explain ( same for line 186-189 and so on…)

Lines 156-276 (including tables 4-9): The information is just mentioned and enumerated… I think you can join all the important information from each table, and create a new one. Also, write 3-4 paragraphs to explain the resistance. How you did it is too hard to follow

Response 9: Thank you. I have changed this paragraph to”Eighteen isolated Pm strains were tested against 21 antibacterial drugs (the numbers of resistant strains, the proportion of drug resistance to total isolates): ampicillin (3 16.67 %), amoxicillin (2, 11.11 %), penicillin (3, 16.67 %), cephalothin (0, 0.00), ceftiofur (1, 5.56 %), cefotaxime (0, 0.00), streptomycin (10, 55.56 %), gentamicin (9, 50 %), amikacin (5, 27.78 %), kanamycin (8, 44.44 %), spectinomycin (1, 5.56 %), apramycin (0, 0.00), erythromycin (1, 5.56%), tilmicosin(14, 77.78 %), doxycycline (0, 0.00), tetracycline (5, 27.78 %), florfenicol (2, 11.11 %), lincomycin (18, 100 %), compound trimethoprim (11, 61.11 %), enrofloxacin (2, 11.11 %), and ciprofloxacin (2, 11.11 %).” Please look at the passage before Table 5-10 in the manuscript. Please see lines 174-181, 195-203, 212-220, 230-238, 251-259 and 269-276.

Reviewer 2 Report

trimoxazole

Minor suggestions if you like to consider

Author Response

Dear Reviewer,

Thank you very much for reviewing our manuscript (vetsci-2495641) and for your valuable comments and suggestions. They are very helpful for improving our manuscript. According to your comments and suggestions, we thoroughly revised our manuscript and submitted it to Antibiotics for further review. The changes based on your suggestions were highlighted in blue colour in the new version. Point by point corrections were made as follows:

In Simple Summary and Abstract,Incomplete sentence, verb is missing! You should improve language. As a suggestion: "Samples were analyzed using methods......." Usually we do not use abbreviations before using the full name.

Response 1: Thank you very much for your comments. According to your suggestions, I changed on to in, added the verb “were collected”, and changed the bacterial disease to its full name. Replace this paragraph with“This study aimed to determine the prevalence of bacterial diseases in pig farms in various regions of Zhejiang Province and surrounding areas. 526 samples were collected from 85 pig farms in Zhejiang Province and surrounding areas. In this study, Samples were analyzed using bacterial isolation and purification, Gram staining, PCR amplification, and antimicrobial susceptibility testing. The isolated bacteria were mainly as follows: Pasteurella multocida (Pm), Bordetella bronchiseptica (Bb), Glasserella parasuis (G. parasuis), Escherichia coli (E. coli), Streptococcus suis (SS), and Actinobacillus pleuropneumoniae (APP).” Please see lines 13-19 and 27-31.

In Introduction, You know that is not right!. You could write that the major porcine bacterial diseases can..... This looks like materials and methods

Response 2: Thank you. I have changed “caused” to “cause” in the manuscript, Please see lines 54. I have changed”Porcine bacterial diseases can be divided into bacterial diseases of the respiratory system and bacterial diseases of the intestinal system.” to “The major porcine bacterial diseases can be divided into bacterial diseases of the respiratory system and bacterial diseases of the intestinal system.” Please see lines 51-53. I have changed the last sentence to”Disc diffusion testing of the isolated bacteria for the detection of drug resistance, data to guide bacterial infection treatment is provided.”Please see lines 66-67.

Tilmicosin is a macrolide antibiotic

Response 3: I'm sorry to make such a mistake. I have added Tilmicosin to macrolide. Please see Table 1. I also made changes in the following table.

You collected 169 samples but analyzed 526. Can you explain better, because as i understand there were originally only 169 samples (or animals?) and they were multiplied taking from the same sample more samples from tissues or organs. How many pigs exist in the sampling area? What is the sampling percentage according the number of raised pigs? Take into consideration the 3 years of sampling period!

The same question about the original number of samples. Do you have a table about for example how many lung samples were collected? You calculated the isolation percentage on the total number of samples (526), which may not be right.

Response 4: 169 refers to 169 batches of samples submitted for examination or epidemiological investigation. For example, a sample sent for inspection may include samples from several pigs or samples from different parts of a pig. So the actual sample size we tested was 526. There are 185 large-scale pig farms in Zhejiang Province, of which 85 were involved in our investigation. Whenever these farms get sick, they send us samples for testing. We also conduct random inspections of pig farms. I have also added in the manuscript. Please see line 124-125.

What do you mean by double and triple infections? For me, this means two or three bacteria involved. If I am right, what were the combinations recorded? What is the outcome of the following percentages you present? Where that adds to the "scientific guidance for the clinical treatment of bacterial diseases" you mentioned?

Response 5: Thank you. You are right, I have shown the mixed infection in the manuscript in the table. Please see table 4.

Line 137, How the percentage was calculated?

Response 6: Thank you. Divide the number of infections by the total number of infections per bacteria.

For example, 36 Pm were detected, single infections were 12, 12÷36=33.33%.

If you sum the numbers of strains you included, there is a difference of 20!!! I am afraid that is not good for the attention you payed writing the article.

Response 7: Thank you. The number of drug susceptibility tests is not the same as the number of bacteria isolated. In the manuscript, the same bacteria in the same batch have almost the same drug resistance, so we only do one drug susceptibility test for the same bacteria in the same batch.

Delete experimental. They are laboratory results, so just results. delete drug. delete the

Response 8: Thank you. I have changed the experimental results to the results, Please see lines 180, 202, 219 and 237. changed drug resistance to resistance, Please see lines 220 and 237. deleted “the”, Please see lines 220 and 237. I have changed the Spaces and punctuation in the manuscript.

Line 193. The ampicillin, penicillin, 191 streptomycin, apramycin, tilmicosin, and compound trimethoprim resistance rates were 192 96.42 %, 96.42 %, 92.86 %, 96.42 %, and 92.86 %, and 96.42 %, respectively.

Response 9: Thank you. I have deleted this sentence. Please see line 204.

In discussion, Was this an experiment or an investigation?

Response 10: Thank you. I have change experiment to investigation. Please see line 336 and 368.

Reviewer 3 Report

In this study, the authors investigated bacterial epidemiological surveys on pig farms in various cities in Zhejiang and some areas in Anhui, and conducted antimicrobial susceptibility testing on the isolated strains.This work provides scientific guidance for the clinical treatment of bacterial diseases. I recommend this for the publication after the authors have addressed the following.

1.Please standardize the format of the figure notes.

2.Line 94: Please confirm the table you referenced.

3.Line 182: Please check the number of isolated Bb strains.

4.Line 201: Please check the number of isolated HPS strains.

5.Line 271-273:Please recheck the values of the resistance rates of APP strains to cephalothin and cefotaxime.

6.Table 9:SS isolate disc diffusion test results?Please confirm the type of strains.

7.Table 9: Replace “0.00” with “0”.

8.Line 280-283:The expression "15 drugs" is inaccurate.Please correct it.

Author Response

Dear Reviewer,

Thank you very much for reviewing our manuscript (vetsci-2495641) and for your valuable comments and suggestions. They are very helpful for improving our manuscript. According to your comments and suggestions, we thoroughly revised our manuscript and submitted it to Antibiotics for further review. The changes based on your suggestions were highlighted in green colour in the new version. Point by point corrections were made as follows:

1.Please standardize the format of the figure notes.

Response 1: Thank you very much for your comments. I have revised all the figures in the manuscript according to the requirements of the magazine

2.Line 94: Please confirm the table you referenced.

Response 2: Thank you. The reference cited for the Hemophilus parahauis pcr primer is incorrect and I have changed it. Please see Table 2.

3.Line 182: Please check the number of isolated Bb strains. 

4.Line 201: Please check the number of isolated HPS strains.

Response 3 and 4: I'm sorry to make such a mistake. The number of bb susceptibility tests was 28 and the number of Hps susceptibility tests was 36. I have changed it in the manuscript. Please see line 187and 204. The number of drug susceptibility tests is not the same as the number of bacteria isolated. In the manuscript, the same bacteria in the same batch have almost the same drug resistance, so we only do one drug susceptibility test for the same bacteria in the same batch.

5.Line 271-273:Please recheck the values of the resistance rates of APP strains to cephalothin and cefotaxime.

Response 5: Thank you. I have corrected the errors in the manuscript. Twenty-eight isolated APP strains were tested against 21 antibacterial drugs(the numbers of resistant strains, the proportion of drug resistance to total isolates): ampicillin (11, 39.29 %), amoxicillin (6, 21.42 %), penicillin (13, 46.42 %), cephalothin (1, 3.57 %), ceftiofur (2, 7.14 %), cefotaxime (0, 0.00), streptomycin (15, 53.57 %), gentamicin (8, 28.57 %), amikacin (15, 53.57 %), kanamycin (15, 53.57 %), spectinomycin (6, 21.42 %), apramycin (24, 85.71 %), erythromycin (3, 10.71 %), tilmicosin(24, 85.71 %), doxycycline (4, 14.28 %), tetracycline (13, 46.42 %), florfenicol (7, 25.00 %), lincomycin (27, 96.43 %), compound trimethoprim (16, 57.14 %), enrofloxacin (11, 39.29 %), and ciprofloxacin (0, 0). Please see line 269-275.

6.Table 9:SS isolate disc diffusion test results?Please confirm the type of strains.

Response 6: Thank you. I've changed the title of table9 to APP isolate disc diffusion test results.

7.Table 9: Replace “0.00” with “0”. 

Response 7: Thank you. I have changed 0 to 0.00 as you requested. Please see Table 9.

8.Line 280-283:The expression "15 drugs" is inaccurate.Please correct it.

Response 8: Thank you. I have changed this sentence to 1 strain was resistant to 12 drugs, accounting for 5.56 %. Please see line 290.

Round 2

Reviewer 1 Report

The authors made extensive improvements to the manuscript, herby, I recommend the publication in our journal.

I suggest carefully reading all the MS again because of some minor grammar mistakes.

Author Response

Dear Reviewer,

Thank you very much for reviewing our manuscript (vetsci-2495641) and for your valuable comments and suggestions. They are very helpful for improving our manuscript. According to your comments and suggestions, we thoroughly revised our manuscript and submitted it to Antibiotics for further review. The changes based on your suggestions were highlighted in purple colour in the new version. Point by point corrections were made as follows:

I suggest carefully reading all the MS again because of some minor grammar mistakes.

Response :Thank you again for reviewing the MS again. According to your valuable suggestion,and modified some mistakes in the manuscript. Please see lines 43. I have changed the comment fonts for Tables 1 and 3. Please see lines74 and 106. And I changed The results revealed that or “The results showed that” to The results reveal that or The results show that in the manuscript. Please see lines 180, 203, 219, 237, 258, 276.

Reviewer 2 Report

I am glad if I offered you any help. The novelty of the work is not importantant but the practical application is very important and i believe it will assist everybody working in the field.

Author Response

Dear Reviewer,

Thank you very much for reviewing our manuscript (vetsci-2495641) and for your valuable comments and suggestions. They are very helpful for improving our manuscript.They are not only very helpful for my manuscript, but also beneficial for my future research path.